# Who Is Willing to Get Vaccinated? A Study into the Psychological, Socio-Demographic, and Cultural Determinants of COVID-19 Vaccination Intentions

**DOI:** 10.3390/vaccines9080810

**Published:** 2021-07-21

**Authors:** Mattia Giuliani, Anna Ichino, Alice Bonomi, Riccardo Martoni, Stefania Cammino, Alessandra Gorini

**Affiliations:** 1IRCCS Centro Cardiologico Monzino, 20138 Milan, Italy; mattia.giuliani@ccfm.it (M.G.); alice.bonomi@ccfm.it (A.B.); 2Department of Philosophy “Piero Martinetti”, University of Milan, 20122 Milan, Italy; anna.ichino@unimi.it; 3Department of Clinical Neurosciences, IRCCS San Raffaele Ville-Turro, 20127 Milan, Italy; martoni.riccardo@hsr.it (R.M.); cammino.stefania@hsr.it (S.C.); 4Department of Oncology and Hemato-Oncology, University of Milan, 20122 Milan, Italy

**Keywords:** COVID-19, COVID-19 vaccination, psychology, motivation, health locus of control, anxiety symptoms, psychological wellbeing

## Abstract

Crucial to the success of the COVID-19 vaccination campaign is the rate of people who adhere to it. This study aimed to investigate the reasons underlying people’s willingness to get vaccinated in a sample of Italian adults, considering the effects of different individual characteristics and psychological variables upon positive vs. negative/hesitant vaccination intentions, as well as subjects’ self-reported motivations for such intentions. An anonymous cross-sectional survey was distributed online in February 2021. The results showed that trust in science, number of vaccinations received in 2019, and belief that COVID-19 is more severe than the common flu, were associated with positive vaccination intentions. “Chance externality” health locus of control showed both direct and indirect effects upon positive vaccination intentions. Anxiety symptoms and participants’ perceived psychological status also showed indirect positive effects. Subjects’ self-reported motivations varied interestingly across positive vs. negative/hesitant intentions. Implications of these findings for identifying effective pro-vaccination messages are discussed in the final section of the paper.

## 1. Introduction

During the last year, scientific laboratories worldwide have worked on an unprecedented timeline [1] to create effective vaccines against the new coronavirus Sars-CoV-2, which is causing the current pandemic [2,3]. In Italy and Europe, 27 December 2020 marked the beginning of the vaccination campaign. This “Vaccine Day” was considered as a symbolic turning point in the management of the state of emergency. Just as vaccines’ availability and supplies are crucial to obtain high vaccination rates, however, so is the population’s willingness to get vaccinated. For this reason, vaccine hesitancy (i.e., the delay in acceptance or refusal of vaccination despite availability of vaccination services) [4,5] was identified by the World Health Organization as one of the top ten global health threats in 2019.

To date, the potential acceptance rates of a generic COVID-19 vaccine as well as the factors influencing acceptance have been investigated in 33 different countries, revealing self-reported acceptance rates that range from almost 90% in China to less than 55% in the Middle East, Russia, Africa, and several European countries [6,7,8,9,10]. Considering that the proportion of the population sufficient to reach herd immunity for COVID-19 is estimated around 70% [11], these data indicate that, in some countries at least, specific interventions are certainly needed to improve vaccine acceptance. Thus, it is crucial and of primary interest for all governments and health institutions to identify factors underlining COVID-19 vaccine hesitancy in order to develop effective strategies to deal with them.

Studies investigating COVID-19 vaccine hesitancy so far have focused primarily on the relationship between sociodemographic factors and cultural and personal beliefs, on the one hand, and the willingness to get vaccinated, on the other. In particular, positive intentions with respect to the COVID-19 vaccination have been shown to be associated with factors such as being male, married, an older adult, having a higher education, being a healthcare professional, having been vaccinated against influenza in the previous season, as well as perceiving a high risk of COVID-19 infection, trusting information from institutional sources, and believing in vaccinations’ efficacy and medical advice more generally [6,9,12,13].

We argue that, in addition to these factors, other variables more closely related to individual psychological characteristics should be taken into account when dealing with vaccine hesitancy, as has been suggested by Rieger [14] and Murphy [15], who found a relation between hesitant/negative vaccination intentions and low altruism, high self-interest, impulsiveness, and a personality characterized by being more disagreeable, more emotionally unstable, and less conscientious.

Building on this, the purpose of the present study was to investigate the effects of different individual characteristics (i.e., sociodemographic status, health condition, and previous decisions about vaccinations), and different psychological variables (such as perceived psychological status, locus of control, and anxiety symptoms) on COVID-19 vaccine acceptance or hesitancy/resistance in a sample of adults in Italy. Moreover, self-reported reasons motivating positive or hesitant/negative vaccination intentions were investigated with the aim of providing insights for public health authorities about what types of incentives or messages are likely to be most effective to increase vaccines community uptake.

## 2. Materials and Methods

### 2.1. Sample

From 6 January to 28 February 2021, an anonymous cross-sectional survey was developed and distributed online across Italy using the Qualtrics software (Provo, UT, USA). The sampling procedure employed was the so-called “Exponential Non-Discriminative Snowball Sampling” [16]. Participants were reached via various social media platforms (i.e., Facebook, LinkedIn, Twitter, and Instagram) and mailing lists, and they were invited to share the survey with their acquaintances. Each participant had the opportunity to “stop and save” the survey and to continue it later. However, once the survey was completed, the link expired, preventing participants from responding more than one time. Inclusion criteria were: (a) age ≥ 18 years and (b) being a native Italian speaker.

A total of 1256 surveys were collected. Of these, 1074 (85.5%) were deemed suitable for the analyses. Of the 182 (14.5%) remaining ones, 166 (91.2%) were excluded because participants completed less than 90% of the survey, and 16 (8.8%) because they did not give their consent to participate (see Figure 1).

The study was approved by the ethical committee of the University of Milan and all respondents included in the analysis signed an online informed consent form before completing the survey.

### 2.2. Survey

The survey collected multiple variables through ad hoc questions similar to those used in other studies [7,8,9,13]. The questions targeted the following issues: (1) sociodemographic factors (i.e., age, sex, education, marital status, region of residence, occupation); (2) participants’ physical status (i.e., perceived physical status, presence/absence of organic diseases); (3) participants’ psychological status (i.e., perceived psychological status, ongoing psychological/psychiatric treatment, symptoms of anxiety, health locus of control); (4) participants’ subjective probability of contracting COVID-19; (5) participants’ fear of contracting COVID-19 (related to self, friends, and family); (6) participants’ trust in government, medical institutions, and science; (7) participants’ opinions about COVID-19’s dangerousness (i.e., “If I contracted COVID-19, my health would be severely damaged”, “The disease caused by COVID-19 is more severe than the common flu”, and “If I went to the hospital, I would probably contract COVID-19”); (8) quantity of vaccinations received in 2019.

Perceived physical and psychological statuses were evaluated on a five-point Likert scale (from 1 = “Poor” to 5 = “Excellent”). Subjective probability of contracting COVID-19 was evaluated through a visual analogue scale ranging from 0 (i.e., “Not likely at all”) to 10 (i.e., “Very likely”). Participants’ fear of contracting COVID-19; their trust in government, health institutions, and science; as well as their opinions about COVID-19’s dangerousness were all evaluated through a five-point Likert scale (from 1 = “Strongly disagree” to 5 = “Strongly agree”). Willingness to get the COVID-19 vaccination was assessed through the following question: “As soon as a COVID-19 vaccine becomes available to you, do you intend to get vaccinated?”. Respondents could answer “Yes”, “No”, or “I do not know” and were then asked to explain their responses (open question). Differently from previous studies that asked subjects to explain their answers only in the case of “No” or “I don’t know” responses (e.g., Fischer et al., 2020), the present study evaluated subjects’ explanations also in the case of “Yes” responses, since the authors were interested in investigating the extent to which the kinds of explanations provided varied across the three different answers. See the Appendix A for the complete version of the Survey and COVID-19 specific questions.

Finally, the present study investigated subjects’ opinions about whether there are any “culprits” to blame for the pandemic. This was done through the following question: “Do you think there is anyone who can be held responsible for the pandemic?”. Subjects who answered “Yes” were then asked to indicate who they did hold responsible. The aim of this question was to identify conspiracist beliefs about the (non-natural—i.e., human, artificial) origins of the virus. 

### 2.3. Symptoms of Anxiety Assessment: The 7-Item Generalized Anxiety Disorder Questionnaire (GAD-7)

All participants were screened for symptoms of anxiety using the GAD-7 [17]. It is a self-report questionnaire involving seven items that adssess the core symptoms of generalized anxiety disorder (GAD) following DSM-IV-TR criteria. Each item score ranges from 0 (“Not at all”) to 3 (“Nearly every day”), with 10 used as a cut-off for clinically relevant symptoms of anxiety. Psychometric evaluations of the GAD-7 suggest that it is a reliable and valid measure of GAD symptoms in both the psychiatric [18,19] and in the general population [20] samples. The GAD-7 has demonstrated good psychometric properties [17]. See Appendix A.

### 2.4. Health Locus of Control Assessment: The Multidimensional Health Locus of Control Scale (MHLCS)

Health locus of control refers to the belief that health is in one’s control (namely, “internal control”) or is not in one’s control (namely, “external control”). Among adults, external locus of control is associated with negative health outcomes, whereas internal locus of control is associated with favorable outcomes [21]. The MHLCS is a self-report questionnaire that evaluates how a person tends to exhibit an “internal” or “external” health locus of control [22]. Specifically, the MHLCS is made of 18 items scored on a five-point Likert scale (1 = “Strongly disagree”, 5 = “Strongly agree”). The MHLCS provides three main subscales: (1) “Internality”, which reflects how much a person believes that his or her own health depends on his or her choices and behaviors; (2) “Powerful Others Externality”, which indicates how much a person believes his or her health depends on other significant people (e.g., family, doctors, partners); (3) “Chance Externality”, which indicates how much a person believes his or her health depends on chance or fate. The MHLCS subscales do not have cut-off points, thus the higher the score, the higher the dimension represented by each subscale (i.e., higher points in the “Internality” subscale indicates higher internal health locus of control). In the present study, only the “Chance externality” subscale was used. See Appendix A.

### 2.5. Willingness to Get COVID-19 Vaccination Motivation: The Categorization Process

Subjects’ self-reported motivations for their willingness/unwillingness to get the COVID-19 vaccination passed through a three-step categorization process. In the first step, independent categorizations were made by three different experimenters. In the second step, the three sets of categories that thereby emerged were compared, and only those categories that had been identified by at least two experimenters were kept. In the third step, those categories went through the external revision of a fourth experimenter. The categories that resulted from this three-step procedure, together with their frequencies and percentages, are listed in Table 1.

Answers containing more than one explanation were sorted into multiple categories (for example, the answer: “(I intend to get the COVID-19 vaccination) because I want to protect myself and the community I live in” was sorted both in the “Self-protection” and in the “Moral/social duty” categories).

### 2.6. Willingness to Get COVID-19 Vaccination: Motivations

Motivations provided for the willingness/unwillingness to get vaccinated against COVID-19 are reported in Table 1. Positive intention to get vaccinated was motivated by the following five main categories of reasons: (1) a social/moral duty to protect one’s community (326 responses); (2) a desire for self-protection (284 responses); (3) a belief in the vaccine’s efficacy (258 responses); (4) a desire to come back to a “normal” (i.e., pre-pandemic) life (127 responses); (5) a general attitude of trust in medical science (126 responses).

Negative or hesitant answers, on the other hand, were motivated by five main sorts of reasons, appealing, respectively, to: (1) concerns about the vaccine’s safety (22 of the “no” responses; 51 of the “I don’t know” responses); (2) concerns about the vaccine’s efficacy (10 of the “no” responses; 14 of the “I don’t know” responses); (3) skepticism about the vaccine’s necessity in relation to the subject’s condition (8 of the “no” responses; 7 of the “I don’t know” responses); (4) personal health issues that make the vaccine specifically contraindicated for the subject (7 of the “no” responses; 11 of the “I don’t know” responses); (5) the existence of effective vaccine alternatives (4 of the “no” responses; 4 of the “I don’t know” responses). In the case of “I don’t know” responses, a further kind of reason provided was: (6) insufficient and confusing information available on the vaccine’s costs and benefits (23 responses). Finally, three respondents reporting a negative intention toward the vaccine explained it by expressing general no-vax attitudes.

Frequencies, percentages and representative examples of the responses belonging to each category are reported in Table 1.

### 2.7. Statistical Analysis

Continuous variables are presented as mean ± standard deviation, and they were compared using a t test for independent samples. Variables not normally distributed are presented as median and interquartile range and were compared with the Wilcoxon rank sum test. Categorical data are reported as frequency and percentage and were compared using an χ2 test or Fisher exact test, as appropriate.

Willingness to get a COVID-19 vaccination was defined as 0 (i.e., “Yes, I will get the vaccination”) and 1 (i.e., “No, I do not intend to get the vaccination” + “I am not sure about my vaccination intentions”). Independent predictors of willingness to get a COVID-19 vaccination were identified via multiple logistic regression analysis with stepwise selection of the variables. The consistency and reliability of the identified subset of predictors were tested by a cross-validation iteration procedure. At each step the dataset was randomly split into two halves. The independent predictors were selected in the first half (training set) and the resulting model was tested for significance in the second half (testing set). The procedure was repeated 200 times with different random splits. The predictor was considered as reliable if it was selected and confirmed at least 75% of the time.

The assessment of direct and indirect effects of psychological variables upon the willingness to get the COVID-19 vaccination was made using path analysis. Health locus of control, anxiety symptoms, and perceived psychological status were used as predictors, and those variables that had been resulting significantly from the cross-validation iteration procedure were used as mediators. The path analysis was performed by using the SAS Proc CALIS procedure (SAS Institute Inc., Cary, NC, USA) based on structural equation modelling. The strength of direct and indirect relationships between variables was quantified by standardized β coefficients.

*p*-values below 0.05 were considered as significant and all tests were two-sided. All analyses were performed using SAS statistical package V. 9.13 (SAS Institute, Inc., Cary, NC, USA).

## 3. Results

### 3.1. Descriptive Statistics

All the variables considered in the study are showed in Table 2.

The mean ages of both groups (respondents who intended to get vaccinated against COVID-19 as well as those who did not or were in doubt) were close to middle age—with a range that varied from 18 to 88 years old. Male prevalence was significantly higher among those who intended to get vaccinated. Marital status did not differ between the two groups. Almost all participants were Italian (i.e., 1069, 99.6%) and most of them were located in Northern Italy. The majority of participants were employed, and the number of healthcare professionals was significantly higher among those who were willing to get vaccinated. Those who intended to receive the COVID-19 vaccine reported a better physical and psychological state compared to the others. Interestingly, those who reported negative or hesitant vaccination intentions had a higher chance externality health locus of control compared to the others. Respondents who intended to receive the COVID-19 vaccine estimated to have a greater chance of contracting the disease and reported greater fear for themselves, their families, and their friends in relation to that possibility. Furthermore, they also reported a higher trust in government, medical institutions, and science.

### 3.2. Is There Anyone Responsible for the COVID-19 Pandemic?

Forty-seven percent of subjects expressing negative/hesitant vaccination intentions said that there is someone responsible for the pandemic, whilst only 27.3% of those with positive vaccination intentions did so. In response to the follow-up open question about who can be held responsible, two main categories of “culprits” were indicated: (1) culprits responsible for the origin of the virus (generally identified as Chinese scientists who supposedly created it in a lab), and culprits responsible for the spread of the virus (identified either as politicians who did not implement effective preventive measures or with the general public who did not respect social distancing).

Among respondents with negative/hesitant vaccination intentions, 73.7% identified the “culprits” as those responsible for the virus origins, while the remaining 26.3% identified the “culprits” as subjects responsible for the virus spread. Among respondents with positive vaccination intentions, 49.5% identified the “culprits” as subjects responsible for the virus origins, and 50.5% as those responsible for the virus spread.

### 3.3. Cross-Validation Procedure

Results of the cross-validation analysis are reported in Figure 2. Variables selected as consistent (percentage of selection > 75%) are listed according to the percentage of selection in the training set. In particular, trust in medical institutions (OR = 0.56; 95% CI = 0.46–0.69) and science (OR = 0.52; 95% CI = 0.40–0.69), the number of vaccinations received in 2019 (OR = 0.25; 95% CI = 0.13–0.49), and the belief that COVID-19 is more severe than the common flu (OR = 0.60; 95%CI = 0.48–0.77) were associated with the positive intention to get vaccinated against COVID-19. Conversely, the variable “significant others’ conflicting opinions about the COVID-19 vaccination” (OR = 2.64; 95% CI = 1.60; 4.37) was associated with negative/hesitant intentions toward the vaccine.

### 3.4. Path Analysis

#### 3.4.1. Multidimensional Health Locus of Control Scale (MHLCS) (Chance Externality)

As shown in Figure 3, the estimated β coefficients showed that MHLCS (chance externality) had both a direct and an indirect effect on the willingness to be vaccinated. Furthermore, it had a direct effect on: (1) the belief that COVID-19 is more severe than the common flu; (2) trust in health institutions; (3) trust in science (Panel A). Notably, the indirect effect of MHLCS (chance externality) on the willingness to be vaccinated passed through trust in health institutions for 68% and through believing that COVID-19 is more severe than the flu for the remaining 32% (Panel B).

#### 3.4.2. Generalized Anxiety Disorder-7 (Symptoms of Anxiety)

As reported in Figure 4 (Panel A), the estimated β coefficients showed that GAD-7 had a direct effect upon: (1) the number of vaccinations received in 2019 by the respondent and (2) significant others’ willingness to be vaccinated against COVID-19, but not upon willingness to be vaccinated. In turn, both the number of vaccinations received in 2019 and the significant others’ willingness to be vaccinated had a direct effect on willingness to be vaccinated.

Moreover, we observed an indirect effect of GAD-7 on the willingness to be vaccinated (Panel B). Interestingly, this relationship passed through the number of vaccinations received in 2019 for 50% and significant others’ willingness to be vaccinated for the remaining 50%.

#### 3.4.3. Perceived Psychological Status

As reported in Figure 5 (Panel A), the estimated β coefficients showed that subjects’ perceived psychological status had a direct effect upon trust in science, number of vaccinations received in 2019, and significant others’ willingness to be vaccinated, but not upon actual willingness to be vaccinated. In turn, the number of vaccinations in 2019 and significant others’ willingness to be vaccinated had a direct effect on the actual willingness to be vaccinated.

There was an indirect effect of perceived psychological status on subjects’ willingness to be vaccinated (Panel B). This effect passed through significant others’ willingness to be vaccinated for 57% and the number of vaccinations performed in 2019 for the remaining 43%.

## 4. Discussion

The main aim of the present study was to investigate which variables may influence the decision to get vaccinated against COVID-19. Results of the cross-validation process showed that the number of vaccinations received in 2019 was the strongest predictor associated with positive COVID-19 vaccination intentions. This result is in line with previous studies showing that a positive intention toward receiving the COVID-19 vaccination is strongly associated with a general tendency to get vaccinated [8,9,13]. Willingness to get vaccinated against COVID-19 was also associated with trust in science and healthcare institutions, again in line with findings from previous studies [23,24]. The third significant factor predicting positive COVID-19 vaccination intentions was the belief that this new virus is more dangerous than the common flu—as indeed it actually is. This result is particularly important, as it indicates that relevant knowledge about a specific disease (in this case, COVID-19) influences subjects’ willingness to be vaccinated against it. This finding also highlights the necessity of clear communication strategies from health and political authorities, especially under pandemic conditions that risks being dominated by confusing or misleading information leading people to engage in risky behavior that can compromise their own others’ health [25].

Interestingly, significant others’ willingness to get COVID-19 vaccination is associated with subjects’ unwillingness to be vaccinated. This finding is quite complex to be explained, but it could be perhaps justified by the fact that the percentage of people who do not intend to get the COVID-19 vaccination is lower than the one of those who want to.

Differently from what was observed in previous studies, which found that older, male, married, and employed subjects with a high income were more favorable toward getting a COVID-19 vaccination [6,7,10,13], no significant association was found between these sociodemographic variables and vaccination preferences. However, the present study results may be explained by the type of analyses performed. In fact, other than running multiple logistic regressions, the authors chose a cross-validation iteration procedure, which has a very strict statistic method. This may have influenced the results, showing only those predictors strongly associated with the study main outcome (i.e., the willingness to get COVID-19 vaccination).

### 4.1. Self-Reported Reasons to Get the COVID-19 Vaccine

The reasons that subjects provided to explain their negative vaccination intentions (“no” responses) turned out to be by and large overlapping with those given to explain hesitant intentions (“I don’t know” responses)—thereby suggesting that, in most cases, the hesitation does not amount to a neutral stance but is closer to a negative one. Both in the case of negative and hesitant vaccination intentions, the most commonly cited reason was a concern about the safety of the vaccine (generally due to lack of sufficient testing) followed by reasons mentioning concerns about the vaccine efficacy (“it is still possible to contract COVID-19 after being vaccinated”) and about the vaccine necessity (“I don’t need the vaccine since I’m not at risk of getting the virus”). In the case of hesitant intentions, another commonly cited reason was the lack of sufficient and clear information on the vaccine’s costs and benefits.

In cases of positive vaccination intentions, on the other hand, the most commonly cited reason was a social/ethical duty to protect one’s community, followed by reasons referring to a desire for self-protection, a belief in the efficacy of the vaccine, a desire to “have one’s life back”, and a general attitude of trust in science.

Importantly, many of these self-reported reasons for positive vs. negative/hesitant vaccination intentions can be seen as the opposite sides of the same coin. Concerns about the vaccine’s safety are clearly the flipside of a desire for self-protection; and skepticism about the vaccine’s efficacy is the flipside of the belief that the vaccine is effective. More generally, the majority of the reasons provided to explain negative and hesitant vaccination intentions presuppose a mistrust in the information from scientific/medical sources, which is the flipside of the trust in science cited by subjects who reported positive vaccination intentions.

This suggests that at least some of the basic desires and needs that are at the root of positive and negative/hesitant vaccination intentions are the same, although subjects have radically different perceptions and beliefs about the world and notably about how such desires and needs can be satisfied (for some subjects, the vaccine, and more generally the solutions indicated by medical authorities, are not suitable means to satisfy their desire to be safe and to find effective ways out of the emergency).

That said, there were also some distinctive reasons that were cited to explain only the positive vaccination intentions and not the negative/hesitant ones (as well as, vice versa, the only negative/hesitant intentions, and not the positive ones). Most notably, the social/ethical considerations that were the most commonly cited reason with which subjects explained their positive intentions to get vaccinated were never mentioned to explain negative/hesitant intentions. This suggests that subjects with positive vaccination intentions perceive themselves as “ethical”—and their own actions as driven by “pro-social” attitudes—much more than subjects with negative/hesitant vaccination intentions do.

Though it is important to note that what is at stake here are indeed self-perceptions: considering that what was collected were self-reported reasons—which might well be influenced by post hoc confabulation ad desirability bias. The findings do not in themselves prove that subjects with positive vaccination intentions are actually driven by ethical and pro-social motivations more than hesitant/no-vaxxers are (also because, of course, insofar as one takes vaccinations to be unsafe and ineffective, refusing them is not unethical from their point of view). Indeed, Rieger (2020) found that when altruistic motivations are triggered in vaccination hesitant/resistant subjects, such motivations do have the potential to influence subjects’ decision-making, eventually leading many of them to shift toward more positive vaccination intentions.

### 4.2. The Role of Beliefs about Human Responsibilities in the Pandemic

Although few participants explicitly mentioned belief in conspiracy theories among the reasons for their negative/hesitant vaccination intentions, previous studies suggest that such beliefs are often associated with (if not directly responsible for) the said intentions. Hornsey et al. found a general connection between conspiratorial thinking and anti-vaccination attitudes [26], and Salali and Uysal found specific associations between beliefs in the human (rather than natural) origin of Sars-CoV-2 and hesitant attitudes toward COVID-19 vaccines [27].

In line with these results, the present study found that positive answers to the question whether there is anyone who can be held responsible for the pandemic were significantly more common among subjects with negative/hesitant vaccination intentions (47%), than among those with positive vaccination intentions (27%). Moreover, even when subjects with positive vaccination intentions said that they thought there were those responsible for the pandemic, often they did not refer to people responsible for the original creation of the virus in a lab, but rather to people responsible for the virus spread (due to failures to adopt adequate preventive behaviors)—which is a rather reasonable view that arguably does not involve any conspiracist claim.

Overall, then, the present study solidly replicates previous findings according to which conspiracist attitudes (and in particular, in this case, conspiracist beliefs about the human origin of the virus) are associated with negative or hesitant vaccination intentions. This suggests that effective information campaigns about the nonhuman origins of the Sars-CoV-2 virus might have positive effects on overcoming vaccine hesitancy.

### 4.3. The Role of Psychological Variables on COVID-19 Vaccine-Related Decisions

The current pandemic is having a significant impact on public mental health. Numerous studies in the last year have pointed out the deleterious effects of the pandemic and the consequent containment measures (i.e., quarantine and social isolation) [28] upon public psychological wellbeing. As has happened in previous epidemics across history [28], an increase in psychological symptoms and disorders in the general population were registered, including, though not limited to, anxiety and depression, stress, feelings of helplessness, anger, and frustration. For this reason, the authors deemed it important to analyze whether and how people’s psychological condition affected their attitudes toward the COVID-19 vaccine.

Quite surprisingly, neither perceived psychological status nor self-reported anxiety had a direct effect on the willingness to be vaccinated. Nevertheless, respondents who reported more anxiety symptoms (in the previous two weeks) and who did not receive any vaccination in 2019 were less willing to receive the COVID-19 vaccine. On the other hand, subjects who reported a generally better psychological status and got a vaccine in 2019 showed more positive intentions to be vaccinated. It might well be that anxiety amplifies doubts and fears about the COVID-19 vaccine, especially in those people who are not that used to getting vaccinations in general, although further studies are needed to test this hypothesis.

Particularly interesting are the results concerning health locus of control. In line with Olagoke et al. [29], higher “chance externality” health locus of control (i.e., assuming that one’s health depends on fate or case) was directly and associated with hesitant or negative vaccination intentions. This is arguably explained by the fact that believing that one’s health does not depend on one’s actions and behaviors is likely to lead one to dismiss COVID-19 vaccination as a useful resource. This result is also in line with another study showing that a higher “chance externality” locus of control was associated with vaccine hesitant/resistant parental attitudes toward child vaccinations [15]. “Chance externality” locus of control had also negative indirect effects on the willingness to get the COVID-19 vaccination, mediated by: (1) trust in science and healthcare institutions, and (2) the belief that COVID-19 can be more severe than the common flu. Again, this might be due to the fact that believing that one’s health depends more on fate than on one’s own actions will lead one to consider healthcare institutions’ advice as irrelevant to deal with COVID-19.

The negative association between chance externality locus of control and the belief that COVID-19 can be more severe than the common flu, on the other hand, may be related to a poor health literacy, as suggested by a previous study [30].

Finally, the fact that negative/hesitant vaccination intentions were associated both with conspiracist beliefs about the human origin of the pandemic and with a higher “chance externality” health locus of control is in line with many previous studies that highlighted a close connection between conspiratorial thinking and high external locus of control in general [31,32].

### 4.4. Study Limitations

This study had the following main limitations. First, despite the attempt to collect data from an Italian geographically distributed sample, the majority of the respondents ended up being from the North of Italy (and in particular from the Lombardy region), so the study sample is not really representative of the entire country and may be biased by the sampling method (i.e., selection bias). Second, most of the respondents turned out to be young and well-educated adults, so the results must be interpreted with caution. Third, the use of a cross-sectional study design makes it hard to establish causality and requires a careful interpretation of results. Fourth, data were collected at the very beginning of the vaccination campaign, so only vaccination intentions and not the actual vaccination uptake could be tested. Moreover, only one vaccine was available at that time, and data about potential lethal side effects related to the COVID-19 vaccine were much more limited than they are now. Fifth, the present study investigated subjects’ self-reported reasons to get vaccinated, which might not (fully) match with the actual reasons that motivated subjects’ responses (which in turn might not always be introspectively available to them). Finally, the motivations to get COVID-19 vaccination categorization process did not employ automated sentiment analysis software, and future studies should implement this qualitative type of analysis.

### 4.5. Conclusions

To date, evidence about psychological factors and personal motivations implicated in the willingness or unwillingness to get the COVID-19 vaccination is limited. The present study started filling this gap based on a sample of Italian respondents, paying close attention to psychological variables—including anxiety symptoms, perceived psychological status, and health locus of control—as well as personal motivations underlying vaccination intentions.

Moreover, the data collected lay the basis for important suggestions about which interventions might prove helpful to increase vaccination acceptance rates. For one thing, the finding concerning the relation between high “chance externality” health locus of control and negative/hesitant vaccination intentions suggests that interventions aimed at boosting people’s sense of control and feelings of empowerment might be effective. Of course, implementing such interventions is not easy. As noted by Van Prooijen (Van Prooijen, 2018), the most promising strategy to boost empowerment is arguably education, the effects of which can only be appreciated in the long-term. But other shorter-term strategies to boost empowerment and perceived control are also possible—such as improving transparency in public decision-making, providing detailed information about the decisions that are enforced upon citizens, as well as giving citizens themselves the opportunity to “voice” their opinions about such decisions, thereby increasing their sense of control about them [33,34].

The findings concerning subjects’ self-reported reasons for their vaccination intentions, on the other hand, provide insights about the sorts of messages that might positively influence such intentions. The fact that concerns about the vaccine’s safety, efficacy, and necessity (together with a lack of sufficient information about the vaccine itself) were the most commonly cited reasons for negative/hesitant vaccination intentions suggests that pro-vaccination messages should seek to address those three sorts of concerns at length, providing extensive and accessible information about each of them. The idea that messages of that sort might positively influence vaccination intentions gets preliminary support from the study by Rieger (Rieger, 2020), which found that a message highlighting that COVID might be dangerous also for young and healthy people, hence that the vaccination might be necessary also for them if they want to avoid health troubles, led a significant amount of hesitant and resistant subjects to change their mind, expressing more favorable vaccination intentions.

In fact, it is worth noting that Rieger’s study showed that even more effective in influencing hesitant and resistant subjects were messages highlighting altruistic reasons in favor of the vaccine—i.e., the sort of “ethical/social” reasons that in the present study were mentioned by respondents with positive vaccination intentions. The fact that those very reasons proved effective to influence also hesitant and resistant subjects makes sense in the light of the present study observation that all subjects, irrespectively of their vaccination intentions, seem to be moved by the same basic desires and needs—whilst they differ in the ways in which they believe those needs can be satisfied. So, perceiving oneself as ethical and altruistic is arguably desirable for everyone, but the point is that some subjects do not see the vaccine as a means to behave altruistically. If it were possible to persuade them that it actually is such a means, this might well make them willing to get vaccinated.

## Figures and Tables

**Figure 1 vaccines-09-00810-f001:**
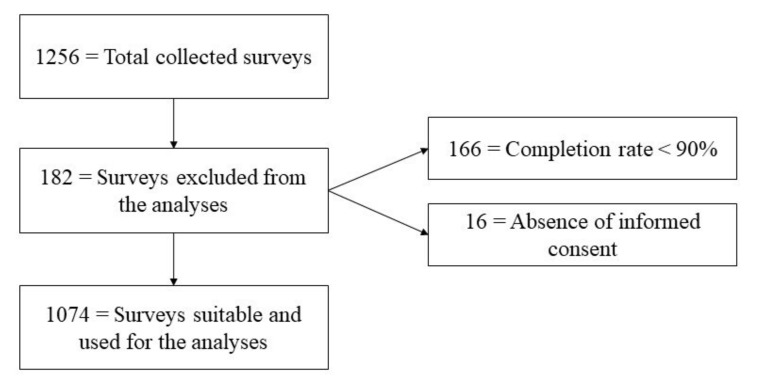
Flow chart.

**Figure 2 vaccines-09-00810-f002:**
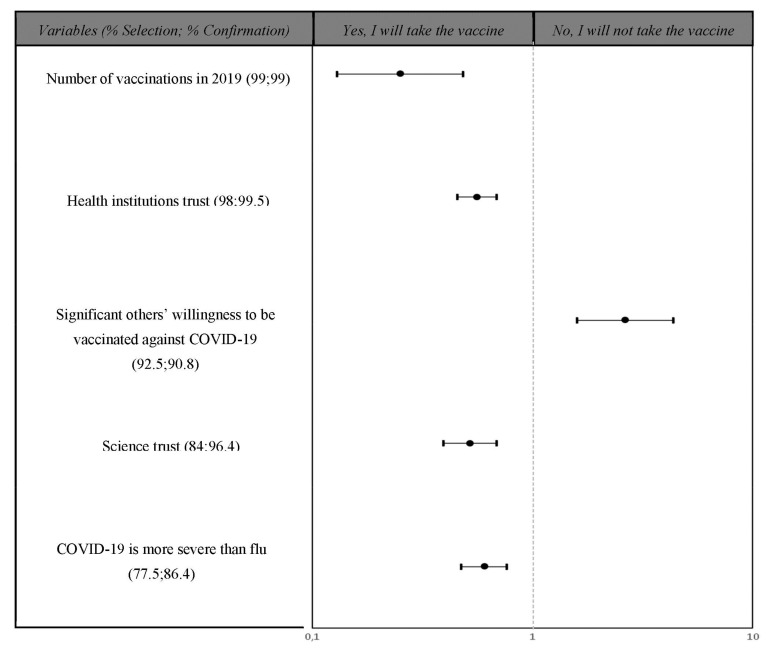
Willingness to get the COVID-19 vaccination predictors according to the cross-validation procedure analysis.

**Figure 3 vaccines-09-00810-f003:**
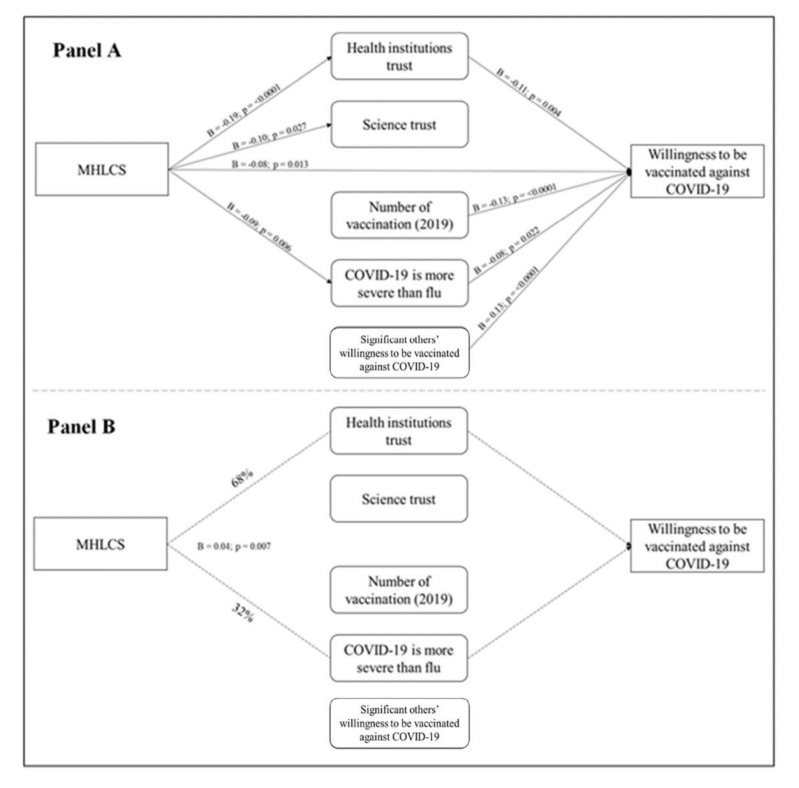
MHLCS (chance externality) path analysis. Panel (**A**) shows the path analysis direct effects: (1) MHLCS on Health institutions trust (*p* = <0.0001); (2) MHLCS on Science trust (*p* = 0.027); (3) MHLCS on Willingness to be vaccinated against COVID-19 (*p* = 0.013); (4) MHLCS on COVID-19 is more severe than flu (*p* = 0.006); (5) Health institutions trust on Willingness to be vaccinated against COVID-19 (*p* = 0.004); (6) Number of vaccination (2019) on Willingness to be vaccinated against COVID-19 (*p* < 0.0001); (7) COVID-19 is more severe than flu on Willingness to be vaccinated against COVID-19 (*p* = 0.022); (8) Significant others’ willingness to be vaccinated against COVID-19 on Willingness to be vaccinated against COVID-19 (*p* < 0.0001). Panel (**B**) shows the path analysis indirect effects (*p* = 0.007).

**Figure 4 vaccines-09-00810-f004:**
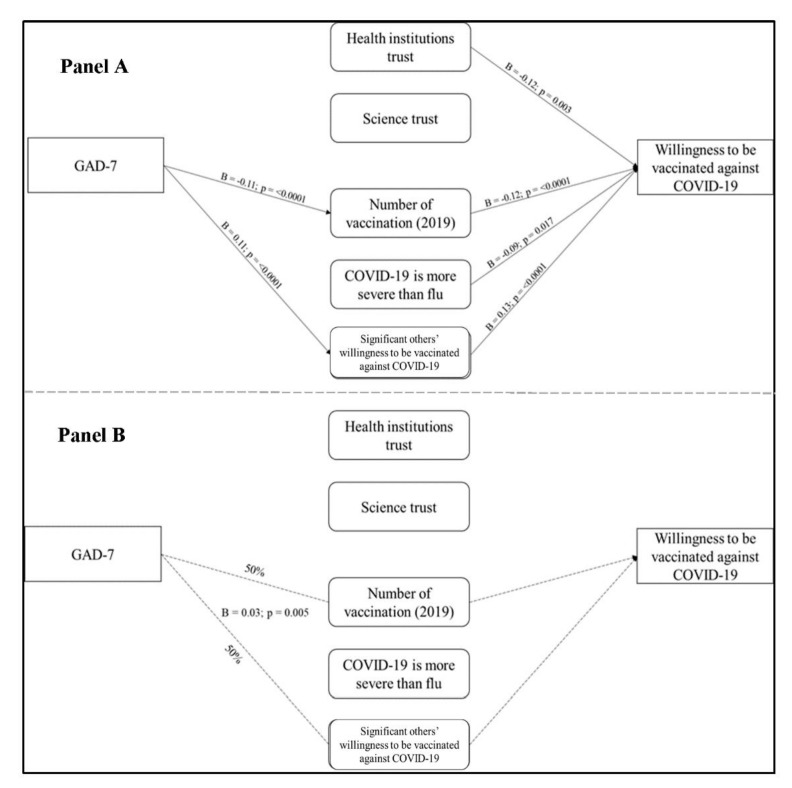
GAD-7 (symptoms of anxiety) path analysis. Panel (**A**) shows the path analysis direct effects: (1) GAD-7 on Number of vaccinations (2019) (*p* = <0.0001); (2) GAD-7 on Significant others’ willingness to be vaccinated against COVID-19 (*p* = <0.0001); (3) Health institutions trust on Willingness to be vaccinated against COVID-19 (*p* = 0.003); (4) Number of vaccination (2019) on Willingness to be vaccinated against COVID-19 (*p* < 0.0001); (5) COVID-19 is more severe than flu on Willingness to be vaccinated against COVID-19 (*p* = 0.017); (6) Significant others’ willingness to be vaccinated against COVID-19 on Willingness to be vaccinated against COVID-19 (*p* < 0.0001). Panel (**B**) shows the path analysis indirect effects (*p* = 0.005).

**Figure 5 vaccines-09-00810-f005:**
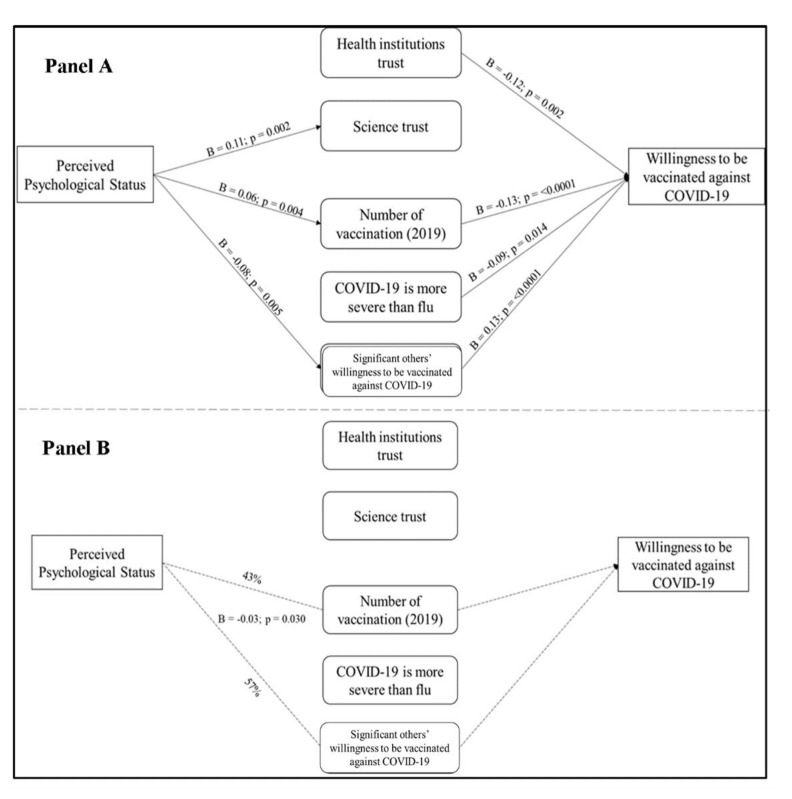
Perceived psychological status path analysis. Panel (**A**) shows the path analysis direct effects: (1) Perceived psychological status on Science trust (*p* = 0.002); (2) Perceived psychological status on Number of vaccinations (2019) (*p* = 0.004); (3) Perceived psychological status on Significant others’ willingness to be vaccinated against COVID-19 (*p* = 0.005); (4) Health institutions trust on Willingness to be vaccinated against COVID-19 (*p* = 0.002); (5) Number of vaccination (2019) on Willingness to be vaccinated against COVID-19 (*p* < 0.0001); (6) COVID-19 is more severe than flu on Willingness to be vaccinated against COVID-19 (*p* = 0.014); (7) Significant others’ willingness to be vaccinated against COVID-19 on Willingness to be vaccinated against COVID-19 (*p* < 0.0001). Panel (**B**) shows the path analysis indirect effects (*p* = 0.030).

**Table 1 vaccines-09-00810-t001:** Motivation categories.

When It Will Become Available to You, Will You Get the COVID-19 Vaccination?	Categories	Examples of the Given Answers	N
Yes, I will	Social/moral duty	It is a social duty to protect my community	326 (28.3%)
	Vaccine’s efficacy	It is the only effective way out of the emergency	258 (22.4%)
	Desire to return to a “normal” (pre-pandemic) life	I want my life back	127 (11.0%)
	Trust	I believe in science	126 (10.9%)
	Self-protection	I do not want to get ill with COVID	284 (27,5%)
No, I will not	Questioning the vaccine efficacy	Apparently, you can catch COVID even if you have been vaccinated	10 (19.6%)
	Personal health issues	I have allergies that make this vaccine unsuitable for me	7 (13.7%)
	Questioning the vaccine’s necessity	I am not at risk of getting COVID	5 (9.8%)
	Generic no-vax opinions	I hate to enrich Big Pharma	3 (5.9%)
	Questioning the vaccine’s safety	The vaccine has not been sufficiently tested; its long-term effects are still unknown	22 (43.1%)
	Alternative remedies available	There are other natural ways to confront COVID-19, such as a healthy lifestyle	4 (7.8%)
I do not know	Questioning the vaccine’s efficacy	I am not sure about the COVID-19 vaccine efficacy	14 (10.3%)
	Personal health issues	I do not know weather the COVID-19 vaccine may worsen my health issues	11 (8.1%)
	Questioning the vaccine’s necessity	I do not know if it is necessary for me	7 (5.1%)
	Questioning the vaccine’s safety	I do not know if the vaccine is safe	51 (37.5%)
	Alternative remedies available	I think that there are other remedies that maybe are useful to prevent COVID-19 infection	4 (2.9%)
	COVID-19 previous infection	I have already contracted COVID-19 and I do not know if I will get the vaccination	3 (2.2%)
	No specific reason	I do not know	2 (1.5%)
	Waiting for medical advice	I will decide after consulting my doctor	7 (5.1%)
	Not decided yet	I have not decided yet	14 (10.3%)
	Insufficient/confusing information	I feel I do ot know enough about this vaccine; I have not received enough information yet	23 (16.9%)

**Table 2 vaccines-09-00810-t002:** Descriptive statistics of the two groups (respondents who intended to get vaccinated against COVID-19 vs. those who did not or were in doubt). Boldface indicates significative *p*-values.

Collected Variables	No + I Do Not Know (*n* = 157; 14.6%)	Yes (*n* = 916; 85.4%)	*p* Value
Age (years)	45.04 ± 13.1	43.9 ± 15.27	0.3794
Sex (men)	37 (23.4)	312 (34.1)	**0.0095**
Education			
Primary school	0 (0)	2 (0.22)	**0.0006**
Secondary school	10 (3.4)	40 (4.4)	
High school	70 (44.6)	277 (30.2)	
University	59 (37.6)	379 (41.4)	
Post-university (e.g., PhD)	18 (11.5)	218 (23.8)	
Marital status			
Single	16 (10.2)	146 (15.9)	0.1244
In a relationship	15 (9.5)	126 (13.7)	
Married	119 (75.8)	601 (65.6)	
Separated/divorced	6 (3.8)	32 (3.5)	
Widowed	1 (0.6)	11 (1.2)	
Geographical origin			
Northern Italy	129 (82.2)	784 (85.6)	0.2521
Center Italy	8 (5.1)	53 (5.8)	
South Italy	20 (12.7)	79 (8.6)	
Occupation (employed)	117 (74.5)	664 (72.5)	0.597
Sanitary job (yes)	34 (21.7)	315 (34.4)	**0.0017**
Perceived physical status	3.87 ± 0.72	4 ± 0.65	**0.0149**
Healthy participants (yes)	113 (71.9)	633 (69.1)	0.4704
Actual diseases number	0.36 ± 0.59	0.40 ± 0.66	0.6160
Perceived psychological status	3.68 ± 0.83	3.82 ± 0.76	**0.0367**
Anxiety problems	39 (24.8)	182 (19.9)	0.1547
Mood problems	23 (14.7)	125 (13.7)	0.7362
Eating problems	11 (7.0)	38 (4.2)	0.1130
Obsession and compulsion problems	5 (3.2)	14 (1.5)	0.1460
Relationship problems	19 (12.1)	103 (11.2)	0.7545
Psychological treatment (yes)	75 (47.8)	414 (45.2)	0.5496
Symptoms of anxiety (GAD-7)	7 ± 5.23	6.13 ± 4.41	0.0998
Health locus of control (MHLCS, chance externality)	15.01 ± 4.68	13.21 ± 4.49	**<0.0001**
Perceived probability of contracting COVID-19 (self)	4.73 ± 2.23	5.21 ± 2.01	**0.0066**
Perceived fear of contracting COVID-19 (self)	3.11 ± 1.14	3.44 ± 0.98	**0.0006**
Perceived fear of contracting COVID-19 (friends)	3.41 ± 1.07	3.75 ± 0.82	**0.0004**
Perceived fear of contracting COVID-19 (family)	3.94 ± 1.02	4.16 ± 0.86	**0.0172**
Perceived probability of severe health damage due to COVID-19	3.25 ± 1.02	3.45 ± 0.97	**0.0204**
Perceived probability of contracting COVID-19 if going to the hospital	2.79 ± 1.08	2.41 ± 1.06	**<0.0001**
Belief that COVID-19 is more severe than the common flu	3.9 ± 0.89	4.37 ± 0.71	**<0.0001**
Government trust	2.68 ± 1.14	3.19 ± 0.99	**<0.0001**
Health institutions trust	2.94 ± 1.06	3.78 ± 0.89	**<0.0001**
Science trust	3.84 ± 0.87	4.48 ± 0.62	**<0.0001**
Significant others’ willingness to be vaccinated against COVID-19	135(86.0)	615 (67.1)	**<0.0001**
Number of vaccinations received in 2019	0 (0;0)	0 (0;1)	**<0.0001**

## Data Availability

All original data are available upon request at IRCCS Centro Cardiologico Monzino, Milan, 20138, Italy.

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
