# Peer review of "Who Is Willing to Get Vaccinated? A Study into the Psychological, Socio-Demographic, and Cultural Determinants of COVID-19 Vaccination Intentions"

_vaccines, 2021, doi:10.3390/vaccines9080810_

Round 1

Reviewer 1 Report

Thank you for inviting me to review this manuscript.The manuscript contains interesting data but it is not so simple to read. Mainly because the concept “Chance externality” health locus of control is not sufficiently explained but also the possibility to have in Appendix the survey questions could help the reader. See below other comments.

2.1 Sample

Authors would  provide more information about how they recruited participants and the eligibility criteria of them. In this paragraph (2.1 Sample) Authors should also provide information about the period of recruitment.

2.2 Survey

In this paragraph Authors should explain with more details the survey questions. Did you use items from validated questionnaires or it was a survey created ad hoc?

Line 129: “Table 1. Motivation categories” contain results and should be reported in Material and Method section the Table 1.

Some questions about participants’ physical status that are reported in the table 2 seem to be very specific, for example Obsessive Compulsive Disorder (OCD) problems or eating disorders. Authors should explain how they collected these type of data.

Furthermore, GAD-7  and locus of control (MHLCS, Chance externality) are  reported in the results but they are not described in this paragraph. Please describe here in the methods section what is it GAD-7 and provide more information about locus of control (MHLCS, Chance externality) otherwise it is difficult to understand the main sense of your work.

Authors would consider to add the survey questions in a supplementary file or appendix.

Results.

Line 170:  the information reported in the sentence “Almost all participants were Italian and most of them were located in Northern Italy” should be reported also in the table 2 or Authors should give more information in the manuscript: how many Italian participants? How many people from the North of Italy? And from the Center and the South?

Lines 186-191: sentences are not clear. Are the absolute numbers referred to the table 1?

Author Response

Thank you for inviting me to review this manuscript.The manuscript contains interesting data but it is not so simple to read. Mainly because the concept “Chance externality” health locus of control is not sufficiently explained but also the possibility to have in Appendix the survey questions could help the reader. See below other comments.

2.1 Sample

Authors would provide more information about how they recruited participants and the eligibility criteria of them. In this paragraph (2.1 Sample) Authors should also provide information about the period of recruitment.

We added the information you have requested in the 2.1 Sample paragraph (lines 77-86).

2.2 Survey

In this paragraph Authors should explain with more details the survey questions. Did you use items from validated questionnaires or it was a survey created ad hoc?

The survey was made up of ad hoc questions. The questions were formulated using previous studies surveys on the same topics as a guide. For further details, see the following papers:

  1. Dror, A. A., Eisenbach, N., Taiber, S., Morozov, N. G., Mizrachi, M., Zigron, A., Srouji, S., & Sela, E. (2020). Vaccine hesitancy: the next challenge in the fight against COVID-19. European Journal of Epidemiology, 35(8), 775–779. https://doi.org/10.1007/s10654-020-00671-y
  2. Fisher, K. A., Bloomstone, S. J., Walder, J., Crawford, S., Fouayzi, H., & Mazor, K. M. (2020). Attitudes Toward a Potential SARS-CoV-2 Vaccine: A Survey of U.S. Adults. Annals of Internal Medicine. https://doi.org/10.7326/m20-3569
  3. Malik, A. A., McFadden, S. A. M., Elharake, J., & Omer, S. B. (2020). Determinants of COVID-19 vaccine acceptance in the US. EClinicalMedicine, 26, 100495. https://doi.org/10.1016/j.eclinm.2020.100495
  4. Wang, J., Jing, R., Lai, X., Zhang, H., Lyu, Y., Knoll, M. D., & Fang, H. (2020). Acceptance of covid-19 vaccination during the covid-19 pandemic in china. Vaccines, 8(3), 1–14. https://doi.org/10.3390/vaccines8030482

Furthermore, the survey was added as an Appendix material.

Line 129: “Table 1. Motivation categories” contain results and should be reported in Material and Method section the Table 1.

We modified the manuscript according to your suggestion (lines 177-198).

Some questions about participants’ physical status that are reported in the table 2 seem to be very specific, for example Obsessive Compulsive Disorder (OCD) problems or eating disorders. Authors should explain how they collected these type of data.

We thank Reviewer 1 for this comment, as it provides us the opportunity to better explain the reasons behind the questions we chose. First, the questions you highlighted are not part of the perceived physical status, but of the perceived psychological status. Given this premise, we asked all the participants if they have ever been treated by a psychologist and/or a psychiatrist (currently or in the past), and, if so, why the reasons why they were treated. This i show we collected the information you have highlighted. All the survey questions are now available in the Appendix.

Furthermore, GAD-7  and locus of control (MHLCS, Chance externality) are  reported in the results but they are not described in this paragraph. Please describe here in the methods section what is it GAD-7 and provide more information about locus of control (MHLCS, Chance externality) otherwise it is difficult to understand the main sense of your work.

Two new paragraphs were added in the Methods section: “2.3. Symptoms of anxiety assessment: the 7-item Generalized Anxiety Disorder questionnaire (GAD-7)” (lines 132-141) and “2.4. Health Locus of Control assessment: the Multidimensional Health Locus of Control Scale (MHLCS)” (lines 143-160), in which we explained the structure of both questionnaires. Furthermore, we added a brief explanation of what health locus of control is in the “health locus of control” paragraph.

Authors would consider to add the survey questions in a supplementary file or appendix.

Thank you for this advice. We have now added the survey as an appendix.

Results.

Line 170:  the information reported in the sentence “Almost all participants were Italian and most of them were located in Northern Italy” should be reported also in the table 2 or Authors should give more information in the manuscript: how many Italian participants? How many people from the North of Italy? And from the Center and the South?

Geographical information were already presented in Table 2 (specifically, “Geographical origin, North, Center and South). However, we decided to add “Italy” in each level of the variables, as it probably was not clear before. Related to the “ethnicity” issue you raised, you are right: the information about the participants origin was not included neither in the text, nor in Table 2. We have now added the percentage of Italian participants (line 235).

Lines 186-191: sentences are not clear. Are the absolute numbers referred to the table 1?

Yes, they are.

Reviewer 2 Report

The paper entitled “Who is Willing to Get Vaccinated? A Study into the Psychological, Socio-Demographic, and Cultural Determinants of COVID-19 Vaccination Intentions” aimed to investigate the reasons underlying people’s willingness to get vaccinated in a sample of Italian adults, considering the effects of different individual characteristics and psychological variables upon positive vs. negative/hesitant vaccination intentions, as well as subjects’ self-reported motivations for such intentions.

The manuscript is well written and well supported by consistent references. The conclusions are largely sound and improve the existing knowledge.

Therefore, I think that the paper has enough quality to be published.

Author Response

The paper entitled “Who is Willing to Get Vaccinated? A Study into the Psychological, Socio-Demographic, and Cultural Determinants of COVID-19 Vaccination Intentions” aimed to investigate the reasons underlying people’s willingness to get vaccinated in a sample of Italian adults, considering the effects of different individual characteristics and psychological variables upon positive vs. negative/hesitant vaccination intentions, as well as subjects’ self-reported motivations for such intentions.

The manuscript is well written and well supported by consistent references. The conclusions are largely sound and improve the existing knowledge.

Therefore, I think that the paper has enough quality to be published.

We thank Reviewer 2 for her/his comments. We are glad to know that you have found our work interesting and able to contribute to deepen the actual knowledge about the relationship between the intentions to get (or not) COVID-19 vaccination and psychological factors.

Reviewer 3 Report

The manuscript presents an investigation on the reasons underlying people’s willingness to get COVID-19 vaccinated in a sample of Italian adults, considering the effects of different individual characteristics and psychological variables upon positive vs. negative/hesitant vaccination intentions, as well as subjects’ self-reported motivations for such intentions. 
Severity of the condition conscience and anxiety symptoms showed to have an indirect positive effect on the willingness to get vaccinated.
The manuscript is well strucutured, the methodology is detailed, and the results support the conclusions, however I have the following concerns:
Since the questionnaire was anonymous, how did the authors prevent respondents from responding more than one time?
Authors should refrain from using personal pronouns such as "we" and "our" throughout the text and I encourage them to write it in an impersonal form of writing.
Reduce the number of self-citations
Did the authors consider using some automated sentiment analysis through using Natural Language Processing for a better quantization of people willingness of taking the vaccine?

Author Response

The manuscript presents an investigation on the reasons underlying people’s willingness to get COVID-19 vaccinated in a sample of Italian adults, considering the effects of different individual characteristics and psychological variables upon positive vs. negative/hesitant vaccination intentions, as well as subjects’ self-reported motivations for such intentions. 
Severity of the condition conscience and anxiety symptoms showed to have an indirect positive effect on the willingness to get vaccinated.

The manuscript is well strucutured, the methodology is detailed, and the results support the conclusions, however I have the following concerns:

Since the questionnaire was anonymous, how did the authors prevent respondents from responding more than one time?

We thank Reviewer 3 for rising this issue, as it was not clearly explained in the Methods section. Qualtrics software provides the opportunity to prevent respondents from responding more than one time with a specific option: once the survey was completed, the link expires. Each participant had the opportunity to stop the survey and finish it later, but when they completed it, the link was no longer working. We added this information in the Methods section, “2.1. Sample” paragraph (lines 82-85).

Authors should refrain from using personal pronouns such as "we" and "our" throughout the text and I encourage them to write it in an impersonal form of writing.

We modified the manuscript as suggested. See lines 71-72, 122-123, 126-127, 214-215, 314, 346, 399-402, 404, 446-448, 465, 474, 487-488, 499, 523-524, 526, 528, 530, 532, 534, 550, 561, 576-577, 583-584.

Reduce the number of self-citations

We modified the manuscript as suggested

Did the authors consider using some automated sentiment analysis through using Natural Language Processing for a better quantization of people willingness of taking the vaccine?

We thank Reviewer 3 for this suggestion. A software capable of doing Natural Language Processing was not available for us this time, but we think it could be very useful for further researches. We added this in the limitations paragraph (lines 537-539).

Round 2

Reviewer 1 Report

I really appreciate Authors' efforts, they have revised the manuscript according to my suggestions. The present version of the manuscript is much better than previous one.